# Customized Fetal Body Mass Index as a Better Predictive Marker for Neonatal Nutritional Status

**DOI:** 10.3390/diagnostics15070877

**Published:** 2025-03-31

**Authors:** Juan Jesús Fernández Alba, María Castillo Lara, José Manuel Jiménez Heras, Jose Diego Santotoribio, Rocío Fuentes Morales, Francisco José Rosa Rubio, Carmen González Macías

**Affiliations:** 1Department of Obstetrics and Gynaecology, University Hospital of Puerto Real, 11-510 Cadiz, Spain; rocfuemor1@gmail.com (R.F.M.); fj.rosarubio@gmail.com (F.J.R.R.); carmengonzalezmacias1973@gmail.com (C.G.M.); 2Institute of Research and Innovation in Biomedical Sciences of the Province of Cadiz (INiBICA), 11-009 Cadiz, Spain; jjimenezheras@yahoo.com (J.M.J.H.); josediego.santotoribio@uca.es (J.D.S.)

**Keywords:** fetal growth, fetal body mass index, birth weight, neonatal nutritional status, fetal customized curves, predictive models

## Abstract

**Background/Objectives:** The diagnosis of fetal nutritional status is of great importance for the accurate evaluation and monitoring of these pregnancies. The objective of the present study is to develop a model that allows for the prenatal assessment of fetal body mass index and to evaluate its diagnostic efficacy in predicting neonatal nutritional status. **Methods:** A retrospective cohort study was conducted to develop and evaluate a new model in the diagnosis of alterations in fetal nutritional status based on the customized fetal body mass index. By establishing the relationship between weight and length, we can calculate the fetal body mass index, which could correlate more effectively with nutritional status. **Results:** A total of 12,633 subjects were recruited, and 9499 were included in our study. Capacities to predict both neonatal malnourishment and overnutrition were calculated for each of the three methods analyzed (BMI, GROW, and IG21st). The receiver operating characteristic curve for each method was developed. The sensitivity and specificity for the assessment of malnutrition were 0.83 and 0.90, respectively. The area under the ROC curve of our method was 0.95 for malnutrition, while for IG21st and GROW, it was 0.80 and 0.79, respectively. **Conclusions:** This study demonstrates a superior diagnostic capacity for alterations in fetal and neonatal nutritional status of this new fetal BMI curve compared to the previously used fetal weight percentile curves.

## 1. Introduction

Fetal growth and fetal growth defects play a crucial role in human development [1]. Normal fetal development during gestation not only indicates a healthy pregnancy but also informs the post-natal and long-term health of neonates [2]. Birth weight has been identified as a strong marker of health risk throughout postnatal life [3].

Fetal growth is influenced by various factors, including hormones, growth factors, gestational conditions, environmental factors, and fetal pathology [4,5,6]. Additionally, maternal diseases occurring during pregnancy, such as diabetes mellitus, hypertensive disorders, and thyroid diseases, can also impact fetal growth [7,8]. Prenatally, the evaluation of fetal nutritional status is predominantly based on the estimation of fetal weight through ultrasound.

Once the fetal weight is estimated by ultrasound, the obtained value is compared with a reference curve. This comparison allows for the classification of the fetus as small, appropriate, or large for its gestational age and sex, depending on the percentile in which it falls in relation to the reference curve used. Currently, there are essentially two types of fetal growth reference curves. The first and most used are empirically created and based on population studies. A good example of this type of curve is INTERGROWTH21st. This standard was created based on the principle that it can be universally applied to all populations, provided the data come exclusively from healthy, well-nourished mothers who experienced a normal pregnancy. The data were collected prospectively in a standardized manner across eight countries. The standard posits that any downward variation in growth and birthweight due to maternal size and ethnic origin indicates stunting and malnutrition and should not be adjusted for [9,10].

The other type of fetal growth curve is the customized one. These curves aim to classify the normality or abnormality of growth by considering the growth potential of the specific fetus being studied. To achieve this, the percentile is calculated by considering certain maternal variables known to influence growth potential, such as height, weight at the beginning of pregnancy, and ethnicity. A good example of this second type is the GROW model, which, in some studies, has shown better performance in identifying small-for-gestational-age fetuses compared to INTERGROWTH21st [11].

After birth, the diagnosis of neonatal nutritional status can be estimated through the integration of various measurements and fetal anthropometric indices, including length, weight, head circumference, and combinations of these, such as the head circumference to length ratio (HC/L) [12], the arm circumference to head circumference ratio, the ponderal index (PI) [13], or the body mass index (BMI). Understanding fetal nutritional status enables early intervention, potentially correcting growth defects and thereby reducing perinatal and neonatal complications [14]. However, existing growth curves, which only consider fetal weight, do not consistently predict fetal and neonatal nutritional status accurately [15]. Our study aims to develop an optimized model that better identifies fetuses with impaired nutritional status. To achieve this, we propose modeling not only fetal weight but also fetal length. By establishing the relationship between these parameters, we can calculate the fetal BMI, which may correlate more effectively with nutritional status. Secondly, we compare the performance of our model with INTERGROWTH21st and GROW, two well-established models frequently utilized in routine clinical practice across numerous countries.

## 2. Materials and Methods

This retrospective cohort study includes all healthy pregnant women whose pregnancies and deliveries were attended at the Obstetrics and Gynecology Department of Puerto Real University Hospital between January 2011 and December 2021. The exclusion criteria applied to pregnancies with chronic hypertension, diabetes mellitus, and thyroid diseases, as well as births with a gestational age of less than 37 weeks or older than 42 weeks.

Statistical analysis:Estimation of the neonatal weight and length at 40th week

We developed a multivariate linear regression model to predict neonatal weight at 40 weeks of gestation, following the methodology outlined in our previous work [16]. To account for potential bias associated with obesity, maternal weight was adjusted for pregnancies where the maternal BMI exceeded 30 kg/m^2^. The correction involved calculating the maternal weight based on a BMI of 30 kg/m^2^. Similarly, maternal weight was adjusted for mothers with a BMI below 18.5 kg/m^2^ by calculating their weight for a BMI of 18.5 kg/m^2^.

Additionally, we developed a multivariate linear regression model to estimate the theoretical length that a specific fetus should reach at 40 weeks of gestation, using maternal variables (age, weight, and height) and fetal sex as predictors.

Estimation of fetal weight and length each week

We calculate the estimated fetal weight at each gestational age using the proportionality curve proposed by Gardosi et al. [17], incorporating the coefficient of variation of fetal weight in our population. The formula is shown in Equation (1).(1)$$\%Fetal\;\;weight=299.1-\left(31.85*GA\right)+\left(1.094*GA^{2}\right)-0.01055\times \left(GA^{3}\right)$$

**Equation (1)**: Weight proportion formula. GA states for gestational age, in weeks.

The length at each week of gestation was calculated using the Abduljalil formula for both male and female fetuses [18] (Equations (2) and (3)).(2)$$HT\;male=0.02+\frac{66.74*{\left(GA-2\right)}^{2.32}}{23^{2.32}+{\left(GA-2\right)}^{2.32}}$$

**Equation (2)**: Height formula for male newborns. GA states for gestational age in weeks.(3)$$HT\;female=0.02+\frac{65.30*{\left(GA-2\right)}^{2.32}}{22.80^{2.32}+{\left(GA-2\right)}^{2.32}}$$

**Equation (3)**: Height formula for female newborns. GA states for gestational age in weeks.

The method we used to estimate fetal length at any gestational age is like the one employed by Gardosi et al. for modeling fetal weight throughout gestation [17]. Firstly, starting from the Abduljalil formula and through a polynomial regression analysis, we developed our own proportionality curve for fetal length (Equations (4) and (5)).(4)$$\%fetal\;height=-71.79+\left(8.06*GA\right)-\left(\;0.12*GA^{2}\right)+0.00056*\left(GA^{3}\right)$$

**Equation (4)**: Length proportion formula for male fetuses. GA states for gestational age in weeks.(5)$$\%fetal\;height=-72.46+\left(8.19*GA\right)-\left(\;0.12*GA^{2}\right)+0.0006*\left(GA^{3}\right)$$

**Equation (5)**: Length proportion formula for female fetuses. GA states for gestational age in weeks.

These formulas allow us to calculate what proportion of the length at 40 weeks of gestation corresponds to each specific gestational age.

By leveraging the theoretical neonatal length predicted by our regression model at 40 weeks of gestation and multiplying this value by the proportion corresponding to a given gestational age, we can calculate the theoretical length that the fetus must have at any gestational age.

Evaluation of the capacity of GROW, INTERGROWTH-21st, and fetal BMI models to predict nutritional status.

In relation to INTERGROWTH-21st (IG21st) and GROW, newborns were classified according to their birth weight as follows:

Small for gestational age (SGA): Birthweight below the 10th percentile.

Adequate for gestational age (AGA): Birthweight between the 10th and 90th percentiles.

Large for gestational age (LGA): Birthweight above the 90th percentile.

We have not found a classification based on fetal or neonatal BMI; so, for this research study, we considered SGA a fetus or newborn whose BMI was under the 10th percentile, appropriate for gestational age when it was between the 10th and 90th percentile, and, finally, large for gestational age when the BMI was above the 90th percentile for the BMI estimated by our customized BMI curve.

Nutritional status at birth was evaluated by calculating the neonatal ponderal index (PI) (Equation (6)).(6)$$Ponderal\;Index=\frac{Weight\;g*100}{{\left(Lenght\;\mathrm{cm}\right)}^{3}}$$

**Equation (6)**: Ponderal index.

When the PI was below the 10th percentile, the newborn was classified as undernutrition. On the other hand, when the PI was above the 90th percentile, the newborn was classified as overnutrition.

In the case of undernutrition, we consider a true positive when the method (customized BMI, GROW, or IG21st) classified the fetus as SGA. On the other hand, regarding overnutrition, we consider a true positive when the method used classified the fetus as LGA.

Subsequently, receiver operating characteristic curves (ROC) were constructed by calculating the true positives, false positives, true negatives, and false negatives for each percentile, with differences between percentiles set at 0.2. Therefore, these values were computed for percentiles ranging from the 99.5th to the 0.5th.

Differences in classifications for SGA and LGA were determined using the chi-square test. Sensitivity, specificity, positive likelihood ratio, and negative likelihood ratio were calculated to assess the nutritional status according to each method. The differences in their predictive capacities for nutritional status were evaluated using the DeLong test.

The statistical analysis of the data was performed with the software R 4.1.3 [19].

All the equations referenced in the study are presented in Appendix A.

## 3. Results

### 3.1. Study Population

We began with a study population of 12,633 pregnancies. After applying the exclusion criteria and removing pregnancies with missing data or input errors, we were left with 9499 pregnancies (Figure 1).

Throughout the sample group, the median neonatal weight was 3282.65 g (±435.19), and the median neonatal length was 50 cm (±3). Among the population studied, 51% were male, and 49% were female. The median maternal age, weight, and height were 32.22 years (±7.16), 62 kg (±12), and 162 cm (±9), respectively. Additionally, the median gestational age in our population was 275 days (±9). All demographic characteristics of the studied population are presented in Table 1.

### 3.2. Customized Fetal BMI Percentile Curve

We developed the customized fetal BMI curve based on the estimated fetal weight and length.

In Table 2, we present the variables and coefficients of our model for estimating fetal weight at 40 weeks of gestation.

We observed that the most significant variable was gestational age, followed by maternal weight (20.61 and 6.71). Oddly enough, we observe that the parity is a statistically significant variable, but its *p*-value is considerably lower than the others.

The predictor variables included in the regression model for estimating theoretical length at 40 weeks of gestation differ slightly from those used to estimate weight at the same gestational age. For instance, in the length prediction model, the variable parity was not statistically significant and was therefore excluded. Conversely, age showed statistical significance and was included in the model. Maternal height was the most influential variable for neonatal length. All predictor variables and their corresponding coefficients are presented in Table 3.

Using the estimated fetal weight and length from the two models mentioned above, we calculated the theoretical fetal BMI at 40 weeks of gestation. Similarly, we calculated the 10th and 90th percentiles for each delivery in our population to classify them as SGA, AGA, or LGA. For both the INTERGROWTH-21st (IG21st) and GROW methods, we calculated percentiles using the coefficient of variance for fetal weight in our population (which was 14%). Additionally, percentiles for our BMI curve were determined using the coefficient of variance for fetal BMI (which was 5.6%). Each method (customized BMI, GROW, and IG21st) resulted in its own classification of SGA and LGA.

This indicates that each of the analyzed methods identifies a different incidence of SGA and LGA. It also highlights that many fetuses classified as normal by GROW and INTERGROWTH21st are classified as small or large when using the fetal BMI. Table 4 illustrates this by comparing the classification of our customized BMI method with the classifications obtained using both GROW and IG21st. The results of the McNemar Chi-square test indicate a *p*-value of less than 2−16 for both IG21st and GROW, confirming that these two methods are statistically different from the BMI method.

### 3.3. Predictive Capabilities

We conducted a comparison of the predictive capabilities for undernutrition among the three methods studied. We observed minimal differences between GROW and IG21st, regardless of the metric used. However, our BMI curve exhibited higher sensitivity (0.83 vs. 0.31–0.30) and diagnostic odds ratio (DOR) (49.41 vs. 6.96–7.84), despite a slight decrease in specificity (0.90 vs. 0.94–0.95). In the evaluation of overnutrition, our BMI curve demonstrated higher sensitivity (0.84 vs. 0.12–0.37) and higher specificity than IG21st (0.91 vs. 0.90), although lower than GROW (0.98). However, the DOR was notably superior in our method (55.29 vs. 6.68–5.16). The summarized results are presented in Table 5 and Table 6.

### 3.4. Comparison of the Predictive Capabilities

For assessing undernutrition, the BMI curve exhibited a statistically higher area under the receiver operating characteristic curve (aucROC) compared to IG21st and GROW (0.95 vs. 0.8–0.79), as shown in Figure 2.

This difference was confirmed by the DeLong test. Similarly, the aucROC for determining overnutrition using the BMI curve was higher (0.95 vs. 0.74–0.74), as demonstrated in Figure 3, and this result was also confirmed by the DeLong test.

## 4. Discussion

The new BMI curve appears to predict neonatal undernutrition more effectively than GROW or IG21st, with a significantly higher sensitivity of 0.83 compared to 0.31 and 0.3 yielded by GROW and IG21st, respectively. Similar results were observed when evaluating LGA fetuses, where our model demonstrated a sensitivity of 0.83, while GROW and IG21st yielded 0.12 and 0.37, respectively, for the estimation of overnutrition. Sensitivity is a valuable metric for evaluating each classification system, as it is particularly relevant for identifying at-risk gestations and because the treatment is noninvasive. However, when comparing different methodologies, it is useful to consider additional metrics that take false positives into account.

For this purpose, we applied the DOR. Our model demonstrated a statistically superior DOR for both undernutrition and overnutrition identification, with values of 49.41 and 55.29, respectively. The closest DOR to this was IG21st for the identification of LGA, which yielded 7.84. The poor capability of GROW to identify overnutrition in our population was unexpected, especially considering that this model is derived from an estimated weight. However, our model calculates weight using linear regression based on anthropometric and clinical variables, while GROW uses a polynomial linear regression based solely on gestational time.

These results show the capabilities of these models when using the traditional 10th percentile to classify SGA and the 90th percentile to classify LGA. Nevertheless, previous work suggests that these weight percentiles are not necessarily optimal, and at least 30% of undergrowth might be expected to fall within the normal weight range [7]. It is therefore appropriate to delve deeper into the analysis and study the identification capabilities of each system at different percentiles. Given the nature of the results and analysis, the most popular method to display these results and the appropriate metric for comparison is to develop an ROC curve. Each point on this curve will show the sensitivity and specificity of each percentile for each method. We decided to create one curve for overnutrition and another for undernutrition to facilitate the comparison of results and interpretation.

The aucROC shows that both IG21st and GROW have similar predictive capabilities for both undernutrition and overnutrition, with values of 0.8 for undernutrition and 0.74 for overnutrition. It also better illustrates that the points around 30% sensitivity are not optimal for these methods, suggesting that these classification methods could benefit significantly from a custom percentile cutoff. However, our customized BMI curve shows a superior aucROC of 0.95 compared to the other methods.

The fact that in our sample the coefficient of variation of BMI is almost three times less than the coefficient of variation of weight could explain, at least in part, the improvement in the aucROC. If we are better able to predict fetal BMI, and this has less variance due to randomness, we are likely to achieve better results, assuming that BMI and weight are both good indicators of nutritional status.

More innovative strategies to assess fetal growth and newborn nutritional status are needed, as it has been established that there is a significant variety of SGA and severe SGA rates using different growth standards [20]. Although there is previous work regarding the determination of nutritional status through SGA, none, to our knowledge, have used ROC curves to evaluate different percentiles of its behavior. A sensitivity of 15% and specificity of 93% are expected, yet we have observed in our population sensitivity of 30% and 95% for methods based on weight.

Recent models utilizing multivariable logistic regression have yielded a sensitivity of 63% and a specificity of 81% [21]. Other authors have achieved better results by customizing the Hadlock weight with a custom coefficient of variance and a custom percentile, which have provided an aucROC of up to 0.864. However, the same authors achieved an aucROC of 0.880 using multivariable logistic regression [15]. Our study provides a superior aucROC of 0.95 using a similar approach of the custom percentile with a custom coefficient of variance but also incorporates a custom estimated BMI.

The main limitation of our study is the criteria we used to establish undernutrition and overnutrition, for which we used ponderal index tables from previous work. We have found several different ways to assess neonatal nutritional status, but none of them have become a gold standard. This lack of a gold standard to establish neonatal nutritional status implies that using other criteria to establish undernutrition or overnutrition, our BMI curve might not demonstrate such superior performance over GROW or IG21st. Furthermore, a future prospective study providing insights into this classification and the performance of our model, as well as an external validation of our model in different populations, would be of interest. Additionally, our study focuses on the performance of using fetal BMI to predict neonatal nutritional status.

The resources available to obstetricians for assessing fetal health are quite limited. Normal growth can serve as a reliable indicator of health. Similarly, growth alterations, whether deficient or excessive, can signal the presence of a pathological process. Currently, growth alteration is primarily based on the estimation of fetal weight, which is complemented by other clinical observations such as the volume of amniotic fluid and the study of uteroplacental circulation using Doppler. These findings inform both the monitoring of the pregnant woman and decision-making processes. We believe that a method with greater sensitivity to detect fetuses with growth alterations could help optimize both monitoring and treatment. In cases of hypertensive stages of pregnancy, increased sensitivity to detect small-for-gestational-age fetuses could lead to earlier interventions. In cases of diabetes, better identification of large-for-gestational-age fetuses could help optimize metabolic control and even determine whether to end the pregnancy. However, these assumptions should be explored in future research, and further studies evaluating the ability of fetal BMI to predict poor perinatal outcomes may be necessary.

## 5. Conclusions

The optimization of the mathematical model, which predicts fetal nutritional status based on an index relating to predicted customized length and fetal weight estimated by ultrasound (fetal BMI), allows for a more accurate assessment of fetal growth alterations compared to other models that rely solely on fetal weight. Given that the new model is based on variables that are routinely recorded in clinical practice, it could be used immediately by any center. We believe that our findings should be corroborated through the conduct of external validation studies based on populations different from ours.

## Figures and Tables

**Figure 1 diagnostics-15-00877-f001:**
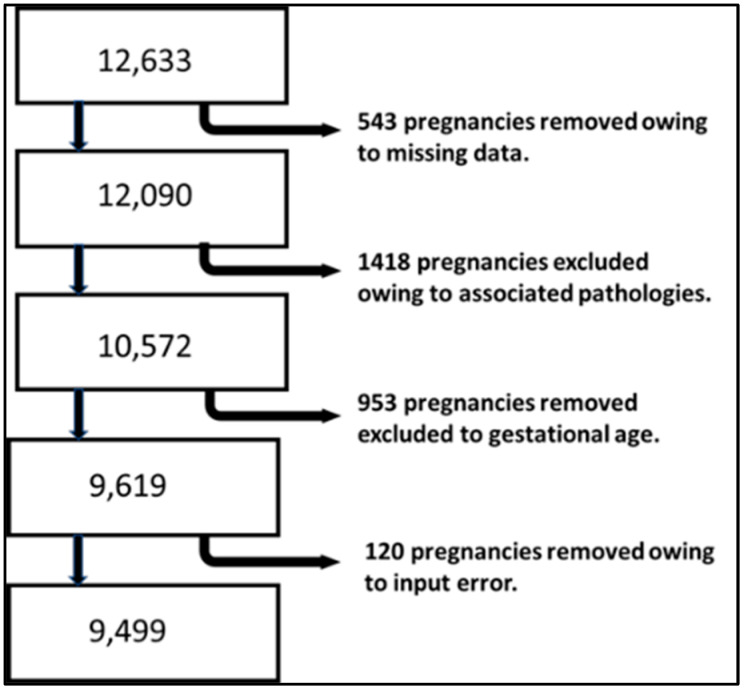
Selection algorithm for the study population.

**Figure 2 diagnostics-15-00877-f002:**
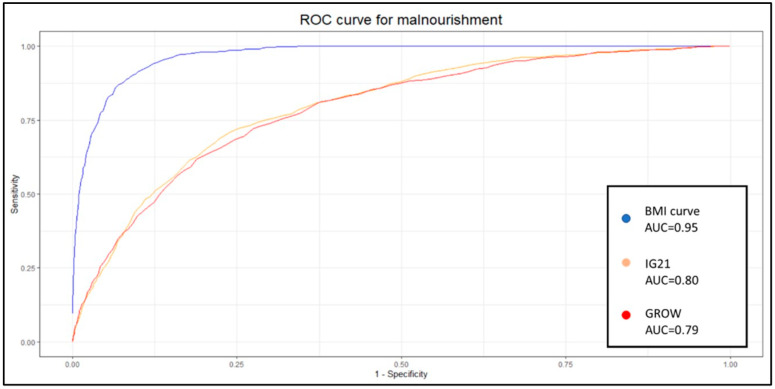
ROC curve for malnourishment for the BMI curve, IG21st, and GROW.

**Figure 3 diagnostics-15-00877-f003:**
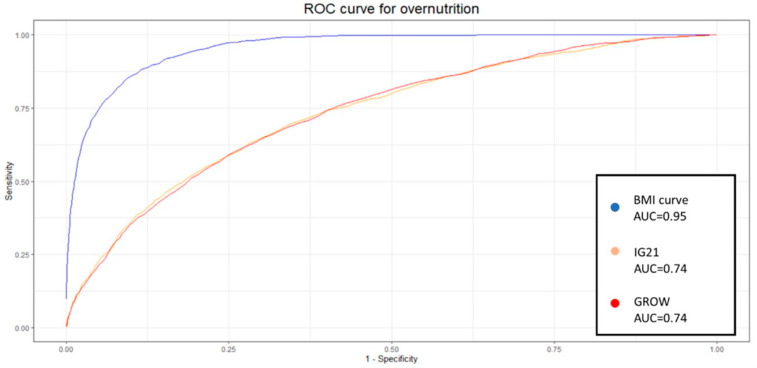
ROC curve for overnutrition for the BMI curve, IG21st, and GROW.

**Table 1 diagnostics-15-00877-t001:** Demographic and clinical characteristics of pregnancies analyzed.

Variable	Median (IQR)/n (%)	
Neonatal weight (g)	3282.65 (435.19)	
Neonatal length (cm)	50 (3)	
Sex	Male	4859 (51%)
	Female	4640 (49%)
Maternal weight (kg)	62 (12)	
Maternal age (years)	32.22 (7.16)	
Maternal height (cm)	162 (9)	
Gestational age(days)	275 (9.8)	
Mode of delivery	Cesarean-section	1914 (20%)
	Vaginal birth	7585 (80%)
Parity	0	5744 (61%)
	1	2973 (31%)
	2	782 (8%)
Gravidity	1	3989 (42%)
	2	4275 (35%)
	3+	2185 (23%
Miscarriage	0	7517 (79%)
	1	1412 (15%)
	2+	570 (6%)

**Table 2 diagnostics-15-00877-t002:** Characteristics of the model for estimated weight.

Variable	Coefficient	Standard Error	*p*-Value
Intercept	1597.43	120.68	<2^−16^
Maternal weight(kg)	6.71	0.58	<2^−16^
Maternal height (cm)	7.59	0.83	<2^−16^
Sex (Male = 1, Female = 0)	121.27	8.72	<2^−16^
Gestational Age (days–280)	20.61	0.51	<2^−16^
Parity	41.19	6.08	1.4^−11^

**Table 3 diagnostics-15-00877-t003:** Characteristics of the model for estimated length.

Variable	Coefficient	Standard Error	*p*-Value
Intercept	41.075	0.59	<2^−16^
Maternal weight (kg)	0.020	0.00	<5.78^−13^
Maternal height (cm)	0.040	0.00	<2^−16^
Sex (Male = 1, Female = 0)	0.791	0.04	<2^−16^
Gestational age (days–280)	0.097	0.00	<2^−16^
Maternal age	0.014	0.00	1.5^−4^

**Table 4 diagnostics-15-00877-t004:** Contingency tables compare the classification of our customized BMI method with the classification obtained using both GROW and IG21st. The tables on the left compare our method with GROW, while the tables on the right compare it with IG21st. The upper tables show the results for SGA, and the lower tables show the results for LGA.

GROW				IG21st		
BMI\GROW	AGA	SGA		BMI\IG21st	AGA	SGA
AGA	7351	118		AGA	7338	122
SGA	1377	653		SGA	1423	602

BMI\GROW	AGA	LGA		BMI\IG21st	AGA	LGA
AGA	7571	47		AGA	7176	442
LGA	1582	299		LGA	991	890

**Table 5 diagnostics-15-00877-t005:** Predictive capabilities for undernutrition of the three methods analyzed.

Undernutrition	BMI Curve	GROW	IG21st
Sensitivity	0.83	0.31	0.30
Specificity	0.90	0.94	0.95
LR+	9.19	5.11	5.81
LR−	0.19	0.73	0.74
DOR	49.41	6.96	7.84

BMI: Body Mass Index; LR+: Positive Likelihood Ratio; LR−: Negative Likelihood Ratio; DOR: Diagnostic Odds Ratio.

**Table 6 diagnostics-15-00877-t006:** Predictive capabilities for overnutrition of the three methods analyzed.

Overnutrition	BMI Curve	GROW	IG21st
Sensitivity	0.84	0.12	0.37
Specificity	0.91	0.98	0.90
LR+	9.27	5.95	3.62
LR−	0.18	0.89	0.70
DOR	55.29	6.68	5.16

BMI: Body Mass Index; LR+: Positive Likelihood Ratio; LR−: Negative Likelihood Ratio; DOR: Diagnostic Odds Ratio.

## Data Availability

The data presented in this study are available on request from the corresponding author [JJFA].

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
