# Peer review of "Customized Fetal Body Mass Index as a Better Predictive Marker for Neonatal Nutritional Status"

_diagnostics, 2025, doi:10.3390/diagnostics15070877_

Round 1
Reviewer 1 Report
Comments and Suggestions for Authors
This is a retrospective cohort study conducted by Fernandez Alba et al, with the aim of developing an optimized model to better identify fetuses with impaired nutrition using a calculated BMI based on fetal weight and length compared to growth charts such as GROW and IG21st. The authors do a nice job at addressing the current limitations encountered with fetal growth and their significance postnatally, but do fail to introduce GROW and IG21st within their introduction. Something must be said as to why they chose those to compare against, as well as their advantages and disadvantages to further justify their aim, method and how that supplements knowledge. It is not until further down the methodology that they introduce the growth charts of comparison, without necessarily explaining "why".
The authors do a nice job in giving enough detail behind their measurements, and as to why they chose to approach the study with the statistics chosen. They also do a nice job at showing the results and in an understandable way, but I do struggle with better understanding what they are trying to show in Table 4. They currently show their method compared to GROW and IG21st in predicting AGA vs SGA, but the numbers don't deliver the message. Are the numbers representing total Ns for one to calculate the Sensitivity and Specificity? More explanation is needed on what the numbers are representing.
It would also be nice to see how the BMIs calculated compare to the z-scores obtained with the IG21st, which is what it primarily uses alongside percentiles.
Also, more can be explained on, or explored with the "so what question?". How is calculating this BMI changing practice prenatally and postnatally? Is it better at predicting postnatal outcomes? i.e more admissions to the NICU, continued poor growth, poor neurodevelopmental outcomes?
I think overall the study was well conducted and presented, with minor revisions needed. The authors explore a novel idea in trying to better predict fetal malnutrition.
Author Response
Dear reviewer
First, we would like to thank you for the effort and time you have dedicated to reviewing our study. We appreciate your kind comments and believe that they all contribute to improving the original manuscript. Below, we will address each of the issues raised.
1.- This is a retrospective cohort study conducted by Fernandez Alba et al, with the aim of developing an optimized model to better identify fetuses with impaired nutrition using a calculated BMI based on fetal weight and length compared to growth charts such as GROW and IG21st. The authors do a nice job at addressing the current limitations encountered with fetal growth and their significance postnatally, but do fail to introduce GROW and IG21st within their introduction. Something must be said as to why they chose those to compare against, as well as their advantages and disadvantages to further justify their aim, method and how that supplements knowledge. It is not until further down the methodology that they introduce the growth charts of comparison, without necessarily explaining "why".
Thank you for the comment. We have expanded the introduction including the next paragraph:
“Once the fetal weight is estimated by ultrasound, the obtained value is compared with a reference curve. This comparison allows for the classification of the fetus as small, appropriate, or large for its gestational age and sex, depending on the percentile in which it falls in relation to the reference curve used. Currently, there are essentially two types of fetal growth reference curves. The first and most used are empirically created and based on population studies. A good example of this type of curve is INTERGROWTH21st. This standard was created based on the principle that it can be universally applied to all populations, provided the data come exclusively from healthy, well-nourished mothers who experienced a normal pregnancy. The data were collected prospectively in a standardized manner across eight countries. The standard posits that any downward variation in growth and birthweight due to maternal size and ethnic origin indicates stunting and malnutrition and should not be adjusted for.
The other type of fetal growth curves are the customized ones. These curves aim to classify the normality or abnormality of growth by considering the growth potential of the specific fetus being studied. To achieve this, the percentile is calculated by considering certain maternal variables known to influence growth potential, such as height, weight at the beginning of pregnancy, and ethnicity. A good example of this second type is the GROW model, which, in some studies, has shown better performance in identifying small-for-gestational-age fetuses compared to INTER-GROWTH21st.”
2.- The authors do a nice job in giving enough detail behind their measurements, and as to why they chose to approach the study with the statistics chosen. They also do a nice job at showing the results and in an understandable way, but I do struggle with better understanding what they are trying to show in Table 4. They currently show their method compared to GROW and IG21st in predicting AGA vs SGA, but the numbers don't deliver the message. Are the numbers representing total Ns for one to calculate the Sensitivity and Specificity? More explanation is needed on what the numbers are representing.
The purpose of these tables is to illustrate that the incidence of SGA and LGA varies according to the method employed. Moreover, as stated in the text, the application of the McNemar Chi-square test reveals that the differences in these incidences are statistically significant, indicating that the three methods classify differently.
We have included a paragraph explaining this question:
“Each method (customized BMI, GROW and IG21st) resulted in its own classification of SGA and LGA. This indicates that each of the analyzed methods identifies a different incidence of SGA and LGA. It also highlights that many fetuses classified as normal by GROW and INTERGROWTH21st are classified as small or large when using the fetal BMI. Table 4 il-lustrates this by comparing the classification of our customized BMI method with the classifications obtained using both GROW and IG21st. The results of the McNemar Chi-square test indicate a p-value of less than 2−16 for both IG21st and GROW, confirming that these two methods are statistically different from the BMI method.”
3.- It would also be nice to see how the BMIs calculated compare to the z-scores obtained with the IG21st, which is what it primarily uses alongside percentiles.
Thank you for the suggestion.
We have not conducted a specific comparison with the z-scores of IG21st, but the values should be similar to those obtained with the percentiles used due to the methodology we employed to calculate these IG21st percentiles. For each individual, the percentile was calculated using the p50 of IG21st as a reference for their gestational age and sex. The formula to calculate it in SPSS is:
Percentile IG21st = CDFNORM (CDFNORM ( ( (weight observed / p50_IG21st) -1 ) * (1 / 0.12) ) * 100
Where 0.12 corresponds to the coefficient of variation of birth weight in our population.
The IG21st percentile for each individual was assigned based on the z-score corresponding to their weight.
4.- Also, more can be explained on, or explored with the "so what question?". How is calculating this BMI changing practice prenatally and postnatally? Is it better at predicting postnatal outcomes? i.e more admissions to the NICU, continued poor growth, poor neurodevelopmental outcomes?
Thank you for the suggestion. It is true that a final paragraph is missing that delves into the potential usefulness of this new method.
We have added the next paragraph at the end of the Discussion section:
“The resources available to obstetricians for assessing fetal health are quite limited. Normal growth can serve as a reliable indicator of health. Similarly, growth alterations, whether deficient or excessive, can signal the presence of a pathological process. Currently, growth alteration is primarily based on the estimation of fetal weight, which is complemented by other clinical observations such as the volume of amniotic fluid and the study of uteroplacental circulation using Doppler. These findings inform both the monitoring of the pregnant woman and decision-making processes. We believe that a method with greater sensitivity to detect fetuses with growth alterations could help optimize both monitoring and treatment. In cases of hypertensive stages of pregnancy, increased sensitivity to detect small-for-gestational-age fetuses could lead to earlier interventions. In cases of diabetes, better identification of large-for-gestational-age fetuses could help optimize metabolic control and even determine whether to end the pregnancy. However, these assumptions should be explored in future research, and further studies evaluating the ability of fetal BMI to predict poor perinatal outcomes may be necessary.”
I think overall the study was well conducted and presented, with minor revisions needed. The authors explore a novel idea in trying to better predict fetal malnutrition.
Thank you very much for your kind comments.
Sincerely
Juan Jesús Fernández Alba

Reviewer 2 Report
Comments and Suggestions for Authors
This is a rather interesting manuscript with intriguing results. I have some issues, which I will point out below:
- I think the terms ”malnourishment” and ”overnutrition” are not adequately used. During fetal life, ”growth” is a more appropriate term that ”nutrition”, so I believe the authors should change that throughout their manuscript. Also, ”malnutrition” can be both ”undernutrition” and ”overnutrition”, if we are completely honest.
- The equations the authors present in the Material and Methods section should be grouped in one comprehensive table, which should also include the formula for the Body Mass Index
- In Table 1, why was the number of miscarriages considered relevant?
- Some of the paragraphs are difficult to comprehend and should be rewritten (lines 151-156).
I included remarks regarding the quality of English in the comments above
Author Response
Dear reviewer
Thank you very much for your valuable comments. We appreciate the time and effort spent reviewing our study. We appreciate your kind comments and believe that they all contribute to improving the original manuscript. Below, we will address each of the issues raised.
This is a rather interesting manuscript with intriguing results. I have some issues, which I will point out below:
1.- I think the terms ”malnourishment” and ”overnutrition” are not adequately used. During fetal life, ”growth” is a more appropriate term that ”nutrition”, so I believe the authors should change that throughout their manuscript. Also, ”malnutrition” can be both ”undernutrition” and ”overnutrition”, if we are completely honest.
Thank you very much for your comment. We have thoroughly reviewed the manuscript and have replaced the terms “malnourishment” and “overnutrition” with more appropriate terminology when referring to the fetus. When these terms are applied to newborns, we have changed both “malnourishment” and “malnutrition” to “undernutrition.”
2.- The equations the authors present in the Material and Methods section should be grouped in one comprehensive table, which should also include the formula for the Body Mass Index
Thank you for the suggestion. We have included a table with all the equations in Appendix A. However, we have also left the equations interspersed in the text as recommended by MDPI in their instructions for authors.
3.- In Table 1, why was the number of miscarriages considered relevant?
Thank you for the comment. The variable “miscarriages” is not of interest to calc the BMI percentile. We only include this variable to adequately describe our sample.
4.- Some of the paragraphs are difficult to comprehend and should be rewritten (lines 151-156).
We have now reviewed the manuscript, particularly the paragraph corresponding to lines 151-156 (196 – 202 in the reviewed manuscript), to enhance its clarity and comprehension.
Thank you very much for your kind comments.
Sincerely
Juan Jesús Fernández Alba
